# SREBP Regulation of Lipid Metabolism in Liver Disease, and Therapeutic Strategies

**DOI:** 10.3390/biomedicines11123280

**Published:** 2023-12-12

**Authors:** Na Li, Xiaodan Li, Yifu Ding, Xiao Liu, Karin Diggle, Tatiana Kisseleva, David A. Brenner

**Affiliations:** 1College of Medical Technology, Shanghai University of Medicine & Health Sciences, Shanghai 201318, China; 2Graduate School of Shanghai University of Traditional Chinese Medicine, Shanghai 201203, China; 3State Key Laboratory of Cell Biology, Shanghai Key Laboratory of Molecular Andrology, CAS Center for Excellence in Molecular Cell Science, Shanghai Institute of Biochemistry and Cell Biology, Shanghai 200031, China; yfding@sibcb.ac.cn; 4Department of Surgery, University of California San Diego, La Jolla, CA 92093, USAtkisseleva@health.ucsd.edu (T.K.); 5Sanford Burnham Prebys, La Jolla, CA 92037, USA

**Keywords:** SREBPs, NAFLD, NASH, fibrosis, hepatocellular carcinoma (HCC), lipogenesis, endoplasmic reticulum (ER) stress, therapeutic target

## Abstract

Sterol regulatory element-binding proteins (SREBPs) are master transcription factors that play a crucial role in regulating genes involved in the biogenesis of cholesterol, fatty acids, and triglycerides. As such, they are implicated in several serious liver diseases, including non-alcoholic fatty liver disease (NAFLD), non-alcoholic steatohepatitis (NASH), fibrosis, and hepatocellular carcinoma (HCC). SREBPs are subject to regulation by multiple cofactors and critical signaling pathways, making them an important target for therapeutic interventions. In this review, we first introduce the structure and activation of SREBPs, before focusing on their function in liver disease. We examine the mechanisms by which SREBPs regulate lipogenesis, explore how alterations in these processes are associated with liver disease, and evaluate potential therapeutic strategies using small molecules, natural products, or herb extracts that target these pathways. Through this analysis, we provide new insights into the versatility and multitargets of SREBPs as factors in the modulation of different physiological stages of liver disease, highlighting their potential targets for therapeutic treatment.

## 1. Introduction

The global burden of nonalcoholic fatty liver disease (NAFLD), now termed metabolic dysfunction-associated liver disease (MASLD), is increasing worldwide in parallel with that of obesity. The prevalence of NAFLD in the overweight population is 79% [1]. NAFLD is characterized by pathologic liver features ranging from steatosis to non-alcoholic steatohepatitis (NASH, now called metabolic dysfunction-associated steatohepatitis, or MASH), characterized by steatosis, inflammation, and fibrosis. Between 20% and 30% of NAFLD cases progress to NASH, and 10–20% of NASH cases progress to cirrhosis (fibrosis stage 4) [2,3]. NAFLD, NASH, and cirrhosis are associated with increased risks of hepatocellular carcinoma (HCC) and liver transplantation [4]. Many lines of evidence have revealed that hepatocyte cell death, inflammation, oxidative stress, endoplasmic reticulum (ER) stress, and lipotoxicity promote MASLD to progress into hepatocarcinogenesis. This review selected and summarized publications on the in vitro and in vivo protein and/or gene expression and function of SREBPs in a range of liver diseases.

## 2. Lipid Metabolism and Liver Diseases: SREBPs Connection

Compared with livers of healthy subjects, de novo lipogenesis (DNL), referring to the endogenous synthesis of lipids (mainly including fatty acid synthesis, fatty acid elongation/unsaturation, and assembly into triglycerides) from dietary sources, is increased by 26% in livers of patients with NAFLD or NASH [5,6,7,8,9], which is a major factor contributing to increased lipid deposition. The alterations in lipid metabolism lead to lipotoxicity, hepatocyte cell death, inflammatory response, and fibrosis, which contribute to NAFL/NASH progression, and further induce hepatocellular stress, injury, and hepatic death, resulting in fibrogenesis and genomic instability and leading to HCC development.

Sterol regulatory element-binding proteins (SREBPs) are transcription factors that regulate lipogenesis, and alterations in this process have been associated with pathogenesis of NAFLD, NASH, and eventually, HCC. SREBPs regulate transcription of genes that control biogenesis of triglycerides (TG), fatty acids (FAs), and cholesterol in mammals [10]. 

### Regulation of SREBPs

Mammalian SREBPs are encoded by the genes *SREBF1* and *SREBF2*. *SREBP1* expresses two isoforms: SREBP-1a and SREBP-1c, which differ in their first exons due to different transcriptional start sites. SREBP1 isoforms are abundant in the liver, white adipose tissue, adrenal glands, skeletal muscle, and brains of mice and humans, while SREBP2 is expressed ubiquitously [11]. SREBP-1c regulates transcription of genes involved in FAs, such as acetyl-CoA carboxylase (ACC) and fatty acid synthase (FAS), two crucial DNL regulation enzymes, and triacylglycerol synthesis, whereas SREBP-1a stimulates both FAs and cholesterol synthesis. SREBP-2 regulates transcription of genes that control cholesterol synthesis. All these pathways interact and overlap in a complicated fashion [12,13]. 

Although SREBPs can be regulated at transcription and protein synthesis levels, most signals target proteolytic activation. The transcription activation domain of SREBPs is located at the N-terminus. SREBPs are expressed as precursors that are cleaved to release the N-terminal domain, which can then enter the nucleus to stimulate the transcription of target genes. An escort protein, SREBP cleavage-activating protein (SCAP), transports SREBP precursors to the Golgi apparatus, where the active forms are generated by two proteases (site 1 protease and site 2 protease) and an anchoring protein. Insulin-induced genes (INSIGs, *Insig1*, and *Insig2*), ER membrane proteins with six transmembrane helices, bind the sterol-sensing domains of SCAP, causing the retention of the SCAP–SREBP complex based on sterol levels [14]. gp78, an ER membrane-anchored E3 ubiquitin ligase identified in sterol-regulated degradation of HMG-CoA reductase (HMGCR), also mediates the degradation of INSIGs proteins, which finally mediates proteolysis of SREBP. Multiple signals, such as sterols, insulin, PCK1, ammonia, etc., can modulate SREBPs’ activation, which further control nuclear SREBPs to regulate the target genes’ expression [15,16,17] (Figure 1).

During development of liver diseases, SREBP activation or repression alters lipid profiles, contributing to metabolic disorders and cancer. Signaling pathways, such as the phosphatidylinositol 3-kinase (PI3K)–protein kinase B (PKB, Akt)–mammalian target of rapamycin (mTOR) pathway, regulate SREBP activation and are altered under different physiologic conditions [18,19]. Strategies to target the SREBP pathways have, therefore, been studied and developed. We review the roles and mechanisms of SREBPs in healthy vs. diseased liver and potential therapies to target these pathways.

## 3. Crucial Roles of SREBPs in Liver Diseases

### 3.1. Multifunctional Role of SREBPs in NAFLD and NASH

Since SREBPs have multifunctional roles in lipid metabolism, many metabolic disorders are related to SREBP deregulation, such as type 2 diabetes, dyslipidemia, atherosclerosis, and hepatic steatosis [20,21]. In NAFLD patients, the rates of lipogenesis are three-fold higher than healthy individuals, and SREBP1c is chronically activated and increases lipogenic activity, resulting in progression of hepatic steatosis [22,23]. Variants of *SREBP*1 also have been associated with increased risk for NAFLD [24]. Transgenic overexpression of SREBP-1c in mouse liver led to increased lipogenesis and hepatic steatosis [25]. On the other hand, deletion of the gene encoding SREBP-1c in livers of *ob/ob* mice, which are insulin resistant, results in an approximate 50% reduction of hepatic triglycerides (TGs), indicating the role of SREBP-1c in the hepatic steatosis in *ob/ob* mice [26]. In *ob/ob* mice, diet-induced NAFLD and steatosis were reversed by SREBP-1c antisense oligonucleotides, without improving insulin hepatic resistance [27]. 

HIV patients with insulin-resistant lipodystrophia have altered hepatic expression of SREBP1 and peroxisome proliferator-activated receptor gamma (PPARγ), compared with NAFLD or control subjects, which contributes to pathogenesis of steatosis and fibrosis [28]. Other virus infections, such as hepatitis B virus (HBV) and hepatitis C virus (HCV), also cause severe liver diseases, but inhibition of the SREBP pathway could repress virus replication, as well as for the dengue virus (DENV) and Zika virus (ZIKV) lifecycles [29,30]. 

Alterations in patatin-like phospholipase3 (PNPLA3) activity have also been associated with development of NASH [31,32]. Polymorphisms in *PNPLA3* have been associated with the severity of NAFLD [33]. Expression of PNPL3 is regulated by SREBP-1c, which binds to the *PNPLA3* promoter. Accumulation of PNPL3 on lipid droplets stimulates lipid accumulation in mouse hepatocytes [33,34].

Insulin mediates the expression of *SREBP1*, proteolytic processing of its products, and thereby, hepatic lipogenesis. Insulin stimulates SREBP-1c to active the PI3K–Akt and Akt–mTORC1 signaling pathways. Furthermore, AMPK, an energy sensor for cellular energy of homeostasis, inhibits cleavage and transcriptional activation of SREBP via phosphorylation. If AMPK activity is stimulated with metformin, SREBP-1c cleavage and nuclear translocation (via ser 372 phosphorylation) are suppressed, leading to attenuation of liver steatosis in mice deficient in low-density lipoprotein (LDL) with diet-induced insulin resistance [35]. 

In patients with NAFLD, treatment with antrodan, which is used to treat alcohol-associated steatohepatitis, alleviates fatty liver symptoms and liver injuries by altering AMPK–Sirt1–PPARγ–SREBP-1c signaling, although the mechanism is not clear [36]. Similarly, xanthohumol inhibits the development of fatty liver in mice by impairing EF–Golgi translocation of the SCAP–SREBP complex and blocking incorporation into common coated protein II vesicles [37]. In patients with hepatic steatosis and lipodystrophy, leptin treatment improves insulin-stimulated hepatic and peripheral glucose metabolism, suggesting potential therapeutic approaches [38]. Recent investigations have shown that 25-HL exhibits both prophylactic and therapeutic effects in alleviating NASH. This is achieved through its binding to INSIG proteins, which in turn stimulates an interaction with SCAP. As a result, SREBP is retained in the endoplasmic reticulum, suppressing its activation [39].

Micro-RNAs (miRNAs) have been identified that regulate lipid metabolism at the post-transcriptional level. The miRNAs can affect intracellular lipid levels, cholesterol transportation, and HDL formation by affecting SREBP expression [40,41]. Long non-coding RNAs (lncRNAs) also regulate hepatic lipogenesis, and alterations in lncRNAs have been associated with NASH. For example, lncARSR is upregulated in serum and liver of NAFLD patients and mouse models by methionine-choline-deficient (MCD) diet feeding. lncARSR mediates lipogenesis via the Akt–SREBP-1c pathway [42]. miRNAs and lncRNAs also regulate lipid metabolism synthetically. In cell lines and mouse models of NAFLD, abundant transcript 1 (NEAT1) binds miR-140 to exacerbate the progression through inactivating the AMPK–SREBP-1 pathway, presenting a new therapeutic strategy [43]. 

Studies support the roles of lncRNAs in hepatic lipid accumulation. For example, the lncRNA Gm15622 is highly expressed in livers of mice with high-fat diet (HFD)-induced obesity. Gm15622 acts as a sponge for the miRNA miR-742-3p, thereby increasing expression of SREBP-1c to promote lipid accumulation. These findings have increased our understanding of how lipid metabolism and accumulation are altered during development of NAFLD [44].

### 3.2. SREBPs Respond to ER Stress-Related Progression of NAFLD to NASH

ER, as a membranous organelle, regulates lipid synthesis. However, nascent protein stimuli and stress disrupt homeostasis, such as genetic mutation, hypoxia, nutrient deficiency, and oxidative stress, leading to the occurrence of endoplasmic reticulum stress (ER stress) [45,46]. During ER stress, the chaperone protein GRP78 (glucose-regulated protein 78), also referred to as BiP, binds to unfolded proteins to cause their dissociation from the luminal domains of the sensors (ATF6, IRE1, and PERK), leading to activation of IRE1a and PERK, by trans-autophosphorylation, and ATF6, by proteolytic processing. This process is referred to as the unfolded protein response (UPR). 

In vitro studies have shown that cholesterol accumulation in the ER leads to activation of SREBP2. In *ob/ob* mice, adenoviral overexpression of BiP in the liver activates SREBP-1c, which reduces ER stress, increases insulin sensitivity, and decreases steatosis. Additionally, SREBP2 processing and its target genes, mainly involving cholesterol metabolism, are also downregulated in the livers of GRP78-expressing *ob/ob* mice, consequently improving hepatic steatosis and dyslipidemia [47,48]. In mice with fatty liver, UPR is usually activated and linked to JNK activation. In transgenic mice that express a constitutively dephosphorylated form of eIF2a in the liver, PPARγ is suppressed, resulting in decreased steatosis [49]. This finding confirms that ER stress signaling is associated with hepatic steatosis.

Hepatocyte apoptosis and inflammation are features of NASH that are linked to the UPR. Fatty acids have been reported to induce apoptosis of steatotic hepatocytes [50]. Overexpression of the chaperone, GRP78, in the livers of *ob/ob* mice reduced markers of ER stress, inhibited cleavage of SREBP-1c, and reduced expression of SREBP-1c and SREBP-2 target genes. These findings might help explain how ER stress contributes to hepatic steatosis in obese individuals [47]. 

A separate study reported that increased levels of uric acid in hepatocyte cells were associated with ER stress induction and hepatic lipid accumulation via activation of SREBP-1c, which promotes lipogenesis through overexpression of the lipogenic enzymes acetyl-CoA carboxylase 1 (ACC1), fatty acid synthase, and stearoyl-CoA desaturase 1 (SCD1) [51]. ER stress promotes the progression of NAFLD to NASH, via the accumulation of hepatocyte-free cholesterol. Caspase-2 co-localizes with site 1 protease (S1P) and cleaves it to generate a soluble active fragment that initiates uncontrolled SCAP-independent activation of SREBP1 and SREBP2, which promotes the buildup of cholesterol and TGs in the liver [52]. 

Agents have been developed to attenuate ER stress, expression of SREBP1 and SREBP2, and reduce liver injury, such as rosiglitazone, naltrexone, and tauroursodeoxycholic acid [53]. Interestingly, ceramide synthases and their derivatives modulate ER stress and NAFLD progression by regulating SREBP-1 cleavage, via decreased levels of INSIG-1, providing another therapeutic approach for NAFLD [54].

### 3.3. SREBPs May Mediate Liver Fibrosis via TGF-β Signaling during Chronic Liver Disease, Progressing to HCC

Liver fibrosis develops with chronic liver disease or injury, such as in NAFLD, hepatitis virus infections, or alcohol-induced steatohepatitis, leading to HCC and liver failure [55]. Liver fibrosis is characterized by an excess deposition of extracellular matrix (ECM) in the perisinusoidal space (between hepatocytes and the sinusoids), distorting the liver architecture and causing liver stiffness [55,56]. The ECM, a network of extracellular macromolecules and minerals, contains enzymes, glycoproteins, and proteins, such as collagen, that provide structural and biochemical support to surrounding cells. Collagen, the main structural protein in connective tissues, is produced by myofibroblasts (hepatic stellate cells in the liver).

Transforming growth factor beta (TGF-β) activates signaling pathways that lead to liver fibrosis. TGF-β signaling, via TβRII and SMADs, causes quiescent HSCs (5–8% of liver cells) to transdifferentiate into activated HSCs, which are highly proliferative, contractile, and fibrogenic. Activated HSCs produce ECM proteins and promote not only fibrogenesis but also hepatic inflammation. TGF-β signaling pathways interact with MAPK, mTOR, PI3K/AKT, and Rho/GTPase pathways [57], and are regulated by SREBP-1c, which inhibits HSC. 

In rat and mouse HSCs, leptin-induced β-catenin signaling reduces the levels of SREBP-1c protein and activity, independent of conventional key regulators of SREBP-1c activity, such as SCAP, INSIG, and other proteins [58]. This observation provides a potential mechanism of the liver fibrogenesis associated with increased levels of leptin in obese individuals and indicates the important function of SREBP signaling in HSCs. 

Further investigation into the mechanisms by which SREBP1c regulates HSCs and liver fibrosis [51] demonstrated that overexpression of SREBP1c inhibited liver fibrosis in mice by reducing the levels of TGFβ1 and signaling via SMAD3 and Akt1/2/3 [59]. In activated HSCs, SREBP-1c was reported to regulate epigenetic factors, such as bromodomain-containing chromatin-modifying factor bromodomain protein 4 (BrD4) and methionine adenosyltransferase 2B (MAT2B), to prevent liver fibrogenesis. Interestingly, PPARγ, a transcription factor that regulates activation of HSCs, increases with levels of SREBP-1c, and together, these proteins inhibit HSC activation. 

SREBP2 has been implicated in liver fibrosis via regulation of cholesterol levels in HSCs. In mouse models of NASH and primary mouse HSCs, the nuclear form of SREBP2 increases with HSC activation. Addition of 25-hydroxycholesterol or LDL, which promotes the formation of the SCAP–INSIG complex, decreased the nuclear form of SREBP2 in quiescent HSCs, but not in activated HSCs. This difference is attributed to differences in the levels of INSIG-1 and INSIG-2 in cells. Activated HSCs have nearly undetectable levels of INSIG-1 and INSIG-2, resulting in the constitutive processing of SREBP when excess free cholesterol accumulates. The feedback regulation of cholesterol homeostasis mediated by SREBP2 is disrupted in HSCs. This disruption affects the sensitivity of HSCs to activation by TGF-ß, resulting in accumulation of free cholesterol and contributing to the vicious cycle of liver fibrosis [60]. Therefore, the SREBP pathway has a crucial role in the process of liver fibrosis and requires careful characterization.

SREBPs are defined as pro-fibrotic mediators because they activate TGF-β in lipotoxicity-induced fibrosis development. There are also investigations into SREBP mediation of kidney fibrosis, via lipid-independent pathways [61]. SREBPs also regulate cell functions, such as autophagy, metabolic circadian rhythm, and other processes that affect expression of non-lipogenic genes in different organs or cell lines during fibrosis development. In light of the similarities among cell and tissue types in fibrosis signaling, it is worthwhile to address the versatility and complexity of SREBPs in mediating liver fibrosis, and hopefully identify inhibitors that can be used to treat liver and other diseases.

### 3.4. Role of SREBPs in HCC

HCC is the second-leading cause of cancer-related death worldwide. Studies have shown that alterations in lipid metabolism and accumulation of lipid metabolic products contribute to hepatocarcinogenesis. HCCs of a high histologic grade and advanced and metastatic tumors overexpress SREBP-1. On the other hand, downregulation of SREBP1 inhibits proliferation and apoptosis in HCC cell lines, such as HepG2 [62]. Tissue microarray analysis showed that high expression of spindlin 1 (SPIN1), associated with HCC malignancy, is co-activated with SREBP-1c, and that the proteins cooperate to increase intracellular TGs, cholesterols, and lipid droplets, which promote progression of HCC [63].

#### 3.4.1. SREBP-1 Expression Level and HCC

SREBP-1 affects HCC development via different signaling pathways. Hepatoma-derived growth factor (HDGF) is co-expressed with SREBP-1 in HCCs, and is closely associated with HCC prognosis [64]. Similarly, apoptosis-antagonizing transcription factors (AATFs) are highly expressed in HCC tissues and cell lines. AATFs bind SREBP-1c to regulate proliferation, migration, colony formation, and tumor growth [65]. 

An HCC-associated tumor suppressor, zinc fingers and homeoboxes2 (ZHX2), inhibits SREBP-1c-regulated lipogenesis in cell lines and human specimens. This regulation pathway induces SREBP-1c degradation by increasing transcription of miRNA-24-3P to suppress HCC development [66]. Acyl CoA synthetase 4 (ACSL4), and oncogene and marker of the alpha-fetoprotein-high subtype of HCC, upregulates expression of SREBP-1 and downstream lipogenic enzymes to reprogram fatty acid metabolism through c-myc, which facilitates HCC progression [67]. In mice with diet-induced obesity that develop steatohepatitis and liver tumors, histone deacetylase 8 (HDAC8) and SREBP-1 are co-expressed in tumor tissues, and HDAC8 is directly upregulated by SREBP-1. Additionally, HDAC8 promotes cell proliferation by coordinating with the chromatin modifier EZH2 to epigenetically repress Wnt antagonists. This repression of Wnt antagonists inhibits cell death through the p53–p21 pathway, leading to G2/M cell cycle arrest. Notably, this regulatory pathway is also perturbed in human NAFLD-associated HCC tissues, in which HDAC8 is aberrantly upregulated by SREBP-1 [68]. 

When signal transducers and activators of transcription 5 (STAT5) are deficient, SREBP-1 and peroxisome PPARγ signaling is upregulated, activating tumor necrosis factor, reactive oxygen species, and STAT3, which contribute to hepatocarcinogenesis [69]. Expression analysis of human HCC specimens showed that SREBP-1 expression correlated with expression of UBC12, which contributes to HCC aggressiveness. If the NEDD8-activating enzyme E1, MLN4924, is incorporated in tumor xenograft mice, HCC progression is blocked [70]. 

mTOR signaling can activate SREBPs. One of the genes regulated by SREBP encodes enzyme fatty acid desaturase 2 (FADS2), which is upregulated in HCC cells [71]. The transcription factor Krüppel-like factor 10 (KLF10) is phosphorylated by AMPK, which then represses SREBP-1c and thereby the lipogenesis pathways in liver tumor cells [72]. 

Another feature of HCC progression is lipidomic alteration, which refers to the rearrangement of lipid components’ structural functions and signal transduction. In NAFLD progression toward HCC, the serum lipidome undergoes significant rearrangement, including a dynamic change in the FA composition. Specifically, saturated fatty acids (SFAs) and polyunsaturated fatty acids (PUFAs) are significantly increased in NAFLD compared to normal livers. These alterations in FA composition are further evident during disease progression: chronic hepatitis -> cirrhosis -> HCC. Many SFAs and MUFAs are increased during disease progression, such as 16:1 for the monounsaturated fatty acids (MUFAs) and 18:1 for the SFAs. However, this is not corroborated in NAFLD-HCC. Despite the slight mechanism difference, the master transcription factor SREBPs predominantly regulate FA biosynthesis pathway key enzymes and directly affect DNL, ultimately mediating various lipid components. Additionally, many oncogene pathways, including the Wnt, P53, and RAS pathways, interplay with SREBPs, resulting in tumorigenesis [73,74,75,76].

#### 3.4.2. SREBPs Cofactors and HCC

Collectively, multiple pathways regulate SREBP-1 expression, activation, and stability to promote the proliferation, invasion, and migration of HCC cells, leading to tumor growth and metastasis. Moreover, genetic disruption or pharmaceutical blocking of regulators of SREBP pathways, such as with SCAP or gp78, inhibits lipid synthesis-related gene expression and reduces HCC progression in chow diet mouse models [77]. In addition to SCAP-dependent SREBP activation during the progression of hepatocellular carcinoma (HCC), SCAP-independent regulation also plays a role. Our research has demonstrated that chronic alcohol consumption-induced HCC upregulates IL-17A signaling, which further increases cholesterol and fatty acid synthesis via TNFR1-caspase-2-dependent activation of SREBP1/2 in both mouse and human steatotic hepatocytes [78]. Deletion of the IL-17A receptor results in significant downregulation of SREBP1/2 proteins and less lipid accumulation in ethanol and HFD-fed mice. Therefore, targeting the IL-17A-SREBP signaling pathway may be a potential therapeutic strategy for patients with alcohol-induced HCC.

#### 3.4.3. SREBP-2 and HCC

SREBP-2 regulates cholesterol synthesis, and SREBP-2 activation promotes cholesterol accumulation that contributes to liver tumor progression. P53 induces the mevalonate pathway through SREBP-2 maturation and induces expression of the cholesterol transporter gene ABCA1. A p53 activator, haplo-insufficient tumor suppressor ASPP2, interacts with SREBP-2 and negatively regulates the mevalonate pathway to inhibit growth of HCC cells [79]. Interaction of the oncogene staphylococcal nuclease with tudor domain-containing-1 (SND-1) leads to altered activation of the sterol regulatory element-binding protein SREBP2 in hepatocellular carcinoma, promoting accumulation of cellular cholesteryl esters [80]. In FASN-knockout mice, which develop liver tumors, levels of HMGCR cholesterol synthesis and nuclear SREBP-2 are increased [81]. If FASN ubiquitination is blocked, the SREBP-1 and SREBP-2 degradation complexes are disrupted, preventing development of liver tumors [82].

## 4. Targeting SREBPs for Treating HCC and Chronic Liver Disease

In targeting SREBPs by modulating the level of expression or the activation process, small molecules or natural products, as potential therapeutics for preventing or treating liver disease, have been thoroughly investigated (Table 1). Song et al. [63] reported that a small molecule, betulin, specifically inhibits the maturation of SREBP by inducing an interaction of SCAP with INSIGs, which could improve hyperlipidemia. The function of betulin in treating liver diseases has been studied in HCC cells. In mice with diethylnitrosamine-induced liver tumors, betulin suppressed tumor progression by reducing inflammation [72]. 

Sorafenib, a multi-kinase inhibitor of RAS–MEK–ERK signaling, VEGFR, and PDGFR, is a first-line treatment for HCC. Sorafenib reduces expression of SCD-1 to decrease synthesis of monounsaturated fatty acids, and activates AMPK to reduce levels of SREBP-1 and phosphorylate mTOR, which suppresses liver cancer [83]. Sorafenib is not an effective treatment for HCC, and high levels of SREBP-1 in tumors correlate with shorter survival. Administration of betulin increases the ability of sorafenib to kill HCC cells and slow the growth of xenograft tumors by suppressing cellular glucose metabolism and reducing glycolytic activity [84].

Radiofrequency ablation is an important strategy for treatment of advanced HCC. After undergoing RFA, tumors in some patients have high protein levels of SREBP-1, which correlates with reduced survival. A small-molecule inhibitor of SREBP-1, SI-1, 1-(4-bromophenyl)-3-(pyridin-3-yl) urea, inhibits aerobic glycolysis, increases killing of HCC cells by radiofrequency ablation, and slows the growth of xenograft tumors. SI-1, 1-(4-bromophenyl)-3-(pyridin-3-yl) urea, therefore, could be a promising approach for treatment of HCC [85]. 

Other compounds, such as cinobufotalin and emodin, inhibit HCC cells via SREBP-1 and its downstream targets. Cinobufotalin, extracted from the skin secretion of the giant toad, promotes HCC cells’ apoptosis, induces cell cycle G2/M arrest, and inhibits cell proliferation. Cinobufotalin downregulates SREBP-1 expression and inhibits de novo lipid synthesis in HCC cells [86]. In addition, emodin, a component of a Chinese medicinal herb Moldenke, induces apoptosis and activates expression of intrinsic apoptosis signaling pathway-related proteins, such as caspase 9, Bax, and BCL-2, in HCC cells. This apoptotic process may or may not depend on the SREBP-1 pathway [87]. Emodin inhibits SREBP-2 transcriptional activity to suppress cholesterol metabolism and Akt signaling, which sensitizes HCC cells to the anti-cancer effect of sorafenib in vitro and in xenograft tumors [88]. Moreover, ursolic acid, a natural pentacyclic terpenoid, activates SREBP-2 and increases the expression of cholesterol biosynthesis-related enzymes to induce cell cycle arrest and apoptosis in HCC cells [89].

**Table 1 biomedicines-11-03280-t001:** Agents that target SREBP-mediated lipogenesis.

Drug	Disease	Mechanism	Targets	Cell Lines Tested	Mouse Models Tested (Dose)	Clinical Trials	Reference
Xanthohumol	Fatty liver	Impairs ER–Golgi translocation of the SCAP–SREBP complex by binding to SEC23 and SEC24 and blocking SCAP–SREBP incorporation into common coated protein II vesicles	SREBP1	Huh-7	HFD-induced fatty liver in male C57BL/6J mice (0.2% or 0.4%)	/	[37]
Antrodan	NAFLD	Reduces HFD-induced NAFLD via the AMPK–SREBP1c–PPARγ pathway	SREBP1	none	(20–40 mg/kg)	/	[36]
Betulin	HCC	Inhibits cell glucose metabolism to prevent metastatic potential and facilitate the inhibitory effect of sorafenib	SREBP1	MHCC97-H	MHCC97-H xenograft tumors (2 mg/kg)	/	[84]
	HCC	Inhibits ER–Golgi translocation of SREBPs	SREBP1	none	Diethylnitrosamine-induced HCC in mice (50 mg/kg)	/	[77]
Emodin	HCC	Induces apoptosis and reduces mitochondrial membrane potential; anti-cancer effects	SREBP1 and its downstream targets, ACLY, ACCa, FASN, and SCD1	Bel-7402	none	/	[87,88]
Sorafenib	HCC	Reduces cell viability	SREBP1 and its target SCD1	human Huh-7.5 liver cancer cells	Huh-7.5 xenograft tumors (20 mg/kg/d)	Phase 4 completed NCT01098760	[83]
Cinobufotalin	HCC	Induces cell cycle G2–M arrest and apoptosis; inhibits cell proliferation by inhibiting de novo lipid synthesis	SREBP1	HepG2, SMMC-7721	SMMC-7721 xenograft tumors (2.5 mg/kg, 5 mg/kg)	Phase 3 NCT03236636	[86]
Ursolic acid	HCC	Activates SREBP2 and cholesterol biosynthesis-related genes and enzymes to lower cholesterol in cells	SREBP2	SK-HEP-1, Hep3B	none	/	[89]

HFD, high-fat diet.

## 5. Conclusions

We have reviewed our understanding of the role of SREBP-mediated lipid metabolism in the development of liver diseases and potential therapeutic targets. Inhibition of SREBP-mediated pathways with small molecules might prevent or slow progression of HCC (Figure 2). Multiple signaling pathways and molecules regulate the expression, stability, and activation of SREBP-1 and SREBP-2, which control gene transcription to regulate liver cancer cell proliferation, apoptosis, endoplasmic reticulum stress, and metastasis (Figure 3). Small molecules, compounds, or herbal extracts (Table 1) that target the SREBP-1- or SREBP-2-regulated mevalonate pathway to repress lipid metabolism might be developed to inhibit liver tumor progression.

However, numerous in vitro and in vivo studies demonstrated SREBPs mediating diverse biological processes, such as glucose homeostasis and hormones synthesis. Specifically, studies have highlighted the potential role of SREBPs in NAFLD, NASH, and HCC. Despite this potential, inhibitors targeting SREBP development for these diseases face many challenges in entering clinical development. An exciting perspective for research would be to determine whether there are cell-type-specific inhibitors to repress the upregulated SREBP pathway under different pathological conditions of liver diseases. Besides, the efficacy should also be considered, as SREBP inhibition is not always beneficial for liver diseases. For example, liver-specific PTEN deficiency results in excess lipids and steatosis in mice, whereas when combined with SCAP deletion it reduces steatosis, exacerbates the later-stage development of inflammation and liver injury, and accelerates the development of NASH and HCC [90]. Targeting lipogenesis via the SREBP pathway is not straightforward and, therefore, alternative approaches that consider lipid levels, based on the physiologic situation, should be explored. A combination of other therapies to enhance efficacy could be another option. Moreover, the safety of SREBP inhibition should be carefully measured, as compensatory pathways and/or toxicity may be evoked after SREBP modulation. In this regard, for different liver pathological stages, such as NAFLD, designing half-life SREBP inhibition strategies to maintain the lipid metabolism at the baseline level could be effective. To effectively address liver pathology, it is crucial to precisely target different stages of disease progression. This approach ensures the maintenance of lipid homeostasis while intervening in lipid metabolism. It remains to be determined whether SREBP inhibition targeting multiple crucial enzymes, cell types, and organ systems could synthesize treatments for diseases beyond liver diseases.

## Figures and Tables

**Figure 1 biomedicines-11-03280-f001:**
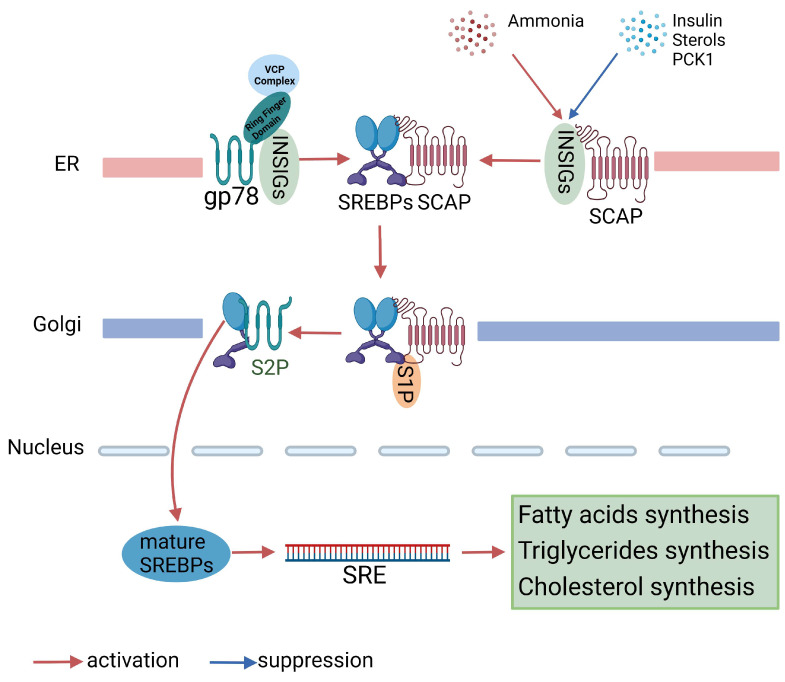
Proteolytic activation of SREBPs. Under basal conditions, SREBPs are complexed with SCAP, INSIGs, and gp78 in the ER membrane to prevent their activation. However, various stimuli, such as sterols, insulin, ammonia, and PCK1, can trigger INSIGs’ degradation, allowing the SREBP-SCAP complex to translocate from the ER to the Golgi apparatus. Within the Golgi, the SREBP is cleaved in a two-step proteolytic mechanism by S1P and S2P. The resulting mature SREBP can then enter the nucleus, where it binds to SRE promoters and activates the transcription of genes involved in fatty acid, triglyceride, and cholesterol synthesis. Please refer to the list of abbreviations for the abbreviations mentioned in this figure.

**Figure 2 biomedicines-11-03280-f002:**
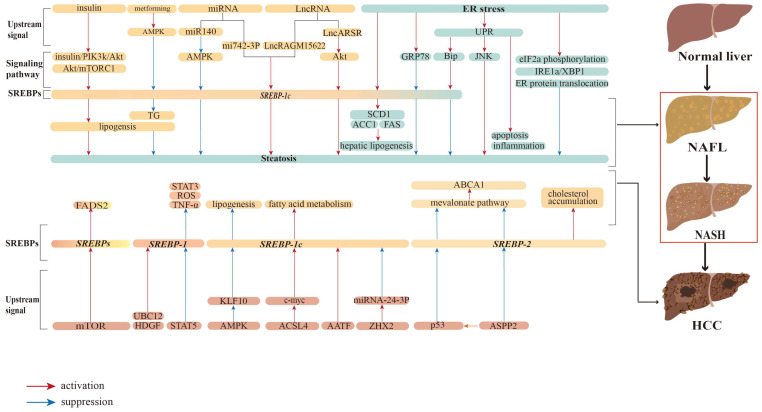
Regulation of the SREBP pathway in liver diseases. Hepatocarcinogenesis involves several stages, progressing through NAFL and NASH. Although progression is slow and partially reversible, the transition from NASH to HCC is irreversible. The SREBP pathway coordinates numerous signaling factors, such as the AKT–mTOR pathway and ER stress. These pathways regulate the different stages of cancer progression. Please refer to the list of abbreviations for the abbreviations mentioned in this figure.

**Figure 3 biomedicines-11-03280-f003:**
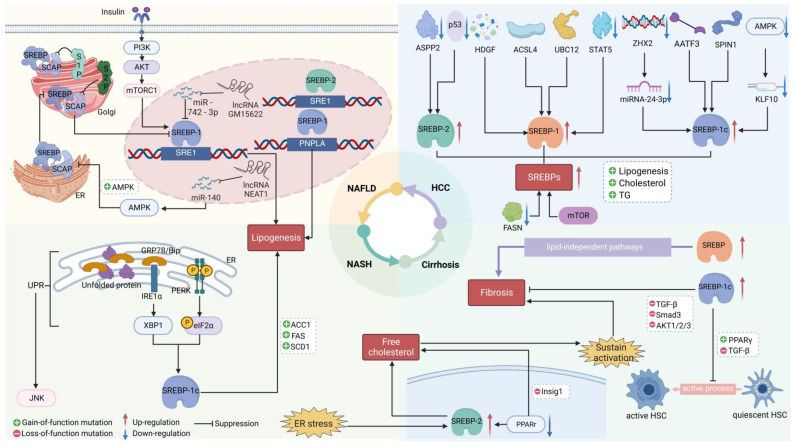
Proper processing of SREBP maintains lipid homeostasis and prevents liver disease. During the development of liver diseases, altered functions of regulators, such as transcription factors, miRNAs, and lcRNAs, affect several steps of lipid metabolism pathways in different cell types. The SREBP pathway is closely intertwined with numerous signaling pathways that regulate many liver functions. ER stress affects lipid synthesis in NAFLD and NASH by modulating the cleavage of SREBP-1c and its downstream target genes, such as FASN, ACC, and SCD-1. INSIGs’ downregulation and ER stress also promote the transcription of SREBP2, leading to an increase in free cholesterol. This increase further promotes the activation of HSCs and contributes to liver fibrosis. SREBPs can also participate in the development of liver fibrosis through the lipid-independent pathway. However, SREBP-1c can inhibit the activation of HSCs and liver fibrosis by inhibiting TGF-β, SMAD3, and AKT. In HCC, various regulatory factors directly or indirectly affect SREBPs, resulting in altered lipid synthesis, cholesterol, and TGs, playing a critical role in the development and progression of HCC. Please refer to the list of abbreviations for the abbreviations mentioned in this figure.

## Data Availability

Not applicable.

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
