# Peer review of "SREBP Regulation of Lipid Metabolism in Liver Disease, and Therapeutic Strategies"

_biomedicines, 2023, doi:10.3390/biomedicines11123280_

Round 1

Reviewer 1 Report

Comments and Suggestions for Authors

I’ve read with attention the paper of Li et al. that is potentially of interest. The background and aim of the study have been clearly defined. The methodology applied is overall correct, the results are reliable and adequately discussed. I’ve only some minor comments:

- The abstract should be enriched with 1-2 short sentences resuming the main/most important/most intringuing new findings on SREBPs

- Even if it is not a systematic review, 1-2 sentences specifying the criteria used to select the references cited in the text should be relevant. 

- The authors should speculate on if safety concerns could be raised by SREBPs modulation

Comments on the Quality of English Language

The paper is ovrerall well-written. No specific concern should be raised on language. 

Author Response

We thank all reviewers for appreciating the quality and value of our work and for all
the excellent suggestions for this review. We revised our manuscript according to these
suggestions. We believe now the paper is much more improved. Below are point-bypoint responses (in red) to each comment. Reviewers’ comments are copied in full in
bold. Changes made in the main text due to the incorporation of new text or discussion
with locations in the main text highlighted in yellow in re-submitted files.
Review1:
Comments and Suggestions for Authors:
I’ve read with attention the paper of Li et al. that is potentially of interest. The
background and aim of the study have been clearly defined. The methodology applied is
overall correct, the results are reliable and adequately discussed.
Response: We are grateful to the reviewer for reviewing our manuscript. We would like to
emphasize that our previous research has also highlighted the vital role of SREBPs in liver
diseases, which complements and reinforces the findings of this study. Thank you for providing
us with the opportunity to clarify this.
I’ve only some minor comments:
- The abstract should be enriched with 1-2 short sentences resuming the main/most
important/most intringuing new findings on SREBPs
Response: Thanks for pointing out this in our abstract. We have reviesed the abstract and
included additional information on line 25 in the updated version.
- Even if it is not a systematic review, 1-2 sentences specifying the criteria used to select
the references cited in the text should be relevant.
Response: Thanks for pointing out this. We add the criteria of selecting the references on line
44 in the updated version.
- The authors should speculate on if safety concerns could be raised by SREBPs
modulation
Response: We agree this comment and addressed the safety concerns on line 390 in the
conclusion section of the updated version.
Comments on the Quality of English Language
The paper is ovrerall well-written. No specific concern should be raised on language.
Response: We thank the reviewer for viewing our manuscript.

Reviewer 2 Report

Comments and Suggestions for Authors

The manuscript  of Na Li et al., is a review concerning the role of SREBP-mediated lipid metabolism in the development of liver diseases and potential therapeutic targets. It reviews the roles and mechanism of SREBPs in healthy versus diseased liver and potential ways to target these pathways.

It is an interesting and  very well written review. Some minor points are included.

Minor points:

- page 2,  line 80 : The authors summarized the role of SREBP in HIV patients. It will be useful to include a small note about other viruses that also are involved in SREBP alterations such as HCV, Zika virus, Dengue virus.

- Figure 1: ER stress should be in bold as steatosis??

- Figure 2: Arrows of up-regulation and down-regulation could be in different colours.

-Please include figure legends not only titles.

Author Response

We thank all reviewers for appreciating the quality and value of our work and for all
the excellent suggestions for this review. We revised our manuscript according to these
suggestions. We believe now the paper is much more improved. Below are point-bypoint responses (in red) to each comment. Reviewers’ comments are copied in full in
bold. Changes made in the main text due to the incorporation of new text or discussion
with locations in the main text highlighted in yellow in re-submitted files.
Review 2:
Comments and Suggestions for Authors
The manuscript of Na Li et al., is a review concerning the role of SREBP-mediated lipid
metabolism in the development of liver diseases and potential therapeutic targets. It
reviews the roles and mechanism of SREBPs in healthy versus diseased liver and
potential ways to target these pathways.
It is an interesting and very well written review. Some minor points are included.
Response: We are grateful to the reviewer for reviewing our manuscript. We would like to
emphasize that our previous research has also highlighted the vital role of SREBPs in liver
diseases, which complements and reinforces the findings of this study.
Minor points:
- page 2, line 80 : The authors summarized the role of SREBP in HIV patients. It will be
useful to include a small note about other viruses that also are involved in SREBP
alterations such as HCV, Zika virus, Dengue virus.
Response: Thanks for pointing out this. We included other viruses involved in SREBP
pathway regulation on line 111 in the updated manuscript.
- Figure 1: ER stress should be in bold as steatosis??
Response: Thank you for bringing to our attention the mistake in the figures. We have
updated the manuscript as per your suggestion and rectified the error. The corrected figure is
now numbered as Figure 2 in the proper sequence.
- Figure 2: Arrows of up-regulation and down-regulation could be in different colours.
Response: Thank you for your valuable suggestion. We have modified the colors to better
differentiate between up and down regulations, making it clearer to readers.
-Please include figure legends not only titles.
Response: Thanks for pointing out this. We have included figure legends as suggestions.

Reviewer 3 Report

Comments and Suggestions for Authors

In this review, Le et al. discuss the role of SREBP-mediated lipid metabolism in the development of liver disease and potential therapeutic targets.

However, the manuscript often lacks clarity and lacks a critical perspective from the Authors, including recommendations for future research.

There are several limitations of this study that should be mentioned.

1. Abstract must be implemented

2. Line 42: “lipid lipogenesis” should be replaced by “lipogenesis"

3. The structure of the review should be restructured in a logical order

- what roles do the SREBPs play in lipid metabolism? (an illustration might be helpful for the reader)

- which molecular mechanisms are responsible for the pathogenesis of non-alcoholic fatty liver disease (NAFLD), non-alcoholic steatohepatitis (NASH), fibrosis and hepatocellular carcinoma (HCC)?

- what are the mechanisms responsible for the activation/repression of SREBPs?

- what role does the activation or repression of SREBPs play in the pathogenesis of NAFLD, NASH, fibrosis, and HCC?

- what are the strategies to target the SREBP pathways?

Author Response

We thank all reviewers for appreciating the quality and value of our work and for all
the excellent suggestions for this review. We revised our manuscript according to these
suggestions. We believe now the paper is much more improved. Below are point-bypoint responses (in red) to each comment. Reviewers’ comments are copied in full in
bold. Changes made in the main text due to the incorporation of new text or discussion
with locations in the main text highlighted in yellow in re-submitted files.
Review 3
In this review, Le et al. discuss the role of SREBP-mediated lipid metabolism in the
development of liver disease and potential therapeutic targets.
However, the manuscript often lacks clarity and lacks a critical perspective from the
Authors, including recommendations for future research.
Response: We have taken your comments into consideration and made the following revisions:
In the conclusion section, we have emphasized the potential therapeutic function of SREBPs
and discussed possible concerns raised by readers about its safety and efficacy. Additionally,
we have added some recommendation on future research investigation that could advance its
clinical transformation. These perspectives are included in the last paragraph of the updated
version.
There are several limitations of this study that should be mentioned.
Response: Thanks for reviewing our manuscript and giving some critique.
1. Abstract must be implemented
Response: Thank you for bringing the issue with the abstract to our attention. We have made
the necessary revisions to the abstract, which now not only summarizes the main investigation
of SREBPs' role in liver diseases but also highlights the most significant conclusion of this
review. We kindly request that you refer to the updated abstract in the revised manuscript. We
believe that the abstract is now much better.
2. Line 42: “lipid lipogenesis” should be replaced by “lipogenesis"
Response: Thanks for pointing out this mistyping and already be replaced as suggested.
3. The structure of the review should be restructured in a logical order
- what roles do the SREBPs play in lipid metabolism? (an illustration might be helpful for
the reader)
- which molecular mechanisms are responsible for the pathogenesis of non-alcoholic fatty
liver disease (NAFLD), non-alcoholic steatohepatitis (NASH), fibrosis and hepatocellular
carcinoma (HCC)?
- what are the mechanisms responsible for the activation/repression of SREBPs?
- what role does the activation or repression of SREBPs play in the pathogenesis of
NAFLD, NASH, fibrosis, and HCC?
- what are the strategies to target the SREBP pathways?
Response: Thanks for providing your suggestion on the review structure. We have made some
modifications to improve the overall quality and clarity.
The introduction of the modified version now emphasizes the significance and importance of
liver diseases on a global scale. We have added a new subtitle to stress the role of master lipid
metabolism regulator SREBPs in liver disease, and the following sections introduce the
molecular structure, general roles, and regulation of SREBPs. An accompanying illustration
(Figure 1) has also been created to assist in understanding this information.
To better address the varied roles of SREBPs in liver disease, we have divided the main text
into sections focused on specific disease types. For each disease, we discuss the specific
mechanisms underlying pathogenesis and modulation activities, as SREBPs activation and
repression mechanisms differ based on disease characteristics. Additionally, we summarize
research investigating the SREBPs pathway as a target for treatment in each disease type.
To improve the logical flow of the review, we have added titles and subtitles (line41, 67, 94,
256, 301, 317) to each section, ensuring that the order of information is clear and organized.
These modifications will enhance the quality of the review overall.

Reviewer 4 Report

Comments and Suggestions for Authors

The authors have tried their best to illustrate the role of SREBP in the Regulation of lipid metabolism in metabolic disorders. The manuscript is written well but the authors need to revise the manuscript as per the suggestions given below:

1. Authors have not included several aspects of the De novo lipogenesis during the progression of Fatty liver disease up to HCC. Describe how the upregulation of SREBP is dominated during NAFLD.

2. How the DNL, unsaturation and elongation of fatty acids is measured? Not covered by the authors. Please incluse the references and description in a paragraph related to this in the manuscript.

3. Are the listed drugs clinically approved?

4. Describe the clinically approved or failure stage of the drugs and also discuss the reason of failure in details.

5. Write down the best strategies to get protected from NAFLD progression in the conclusion section.

Author Response

We thank all reviewers for appreciating the quality and value of our work and for all
the excellent suggestions for this review. We revised our manuscript according to these
suggestions. We believe now the paper is much more improved. Below are point-bypoint responses (in red) to each comment. Reviewers’ comments are copied in full in
bold. Changes made in the main text due to the incorporation of new text or discussion
with locations in the main text highlighted in yellow in re-submitted files.
Review 4
The authors have tried their best to illustrate the role of SREBP in the Regulation of lipid
metabolism in metabolic disorders. The manuscript is written well but the authors need
to revise the manuscript as per the suggestions given below:
1. Authors have not included several aspects of the De novo lipogenesis during the
progression of Fatty liver disease up to HCC. Describe how the upregulation of SREBP is
dominated during NAFLD.
Response: Thanks for bringing attention to this important matter. The role of de novo
lipogenesis (DNL) in the progression of non-alcoholic fatty liver disease (NAFLD) to
hepatocellular carcinoma (HCC) is of great significance. Because SREBPs are master
regulators of DNL, it is important to address this issue. As your suggestion, we have included
a detailed explanation and relevant studies in the revised manuscript to fully elucidate the role
of SREBPs in lipogenesis. These enhancements can be found within lines 47-53 and have
significantly improved the overall quality and clarity of the manuscript.
2. How the DNL, unsaturation and elongation of fatty acids is measured? Not covered by
the authors. Please incluse the references and description in a paragraph related to this
in the manuscript.
Response: Thanks for pointing out this. The alteration of lipidomic is a crucial feature that
contributes to the progression of liver diseases. To enhance the manuscript's content, we have
incorporated additional relevant references and have summarized the investigation in line 287-
299 of the revised version. This improved section highlights the significance and intricacy of
SREBPs' functionality.
3. Are the listed drugs clinically approved?
Response: Thanks for pointing out this. Table 1 includes a list of potential agents that target
SREBPs mediated pathways. It is worth noting that the majority of these agents are not yet
clinically approved, with only a handful currently undergoing clinical trials as documented by
Pubmed. To address this gap in information, I have added a new column to Table 1 that provides
details on ongoing clinical trials. By incorporating this information, there is potential to enhance
the understanding of these potential drugs and their efficacy.
4. Describe the clinically approved or failure stage of the drugs and also discuss the reason
of failure in details.
Response: Thank you for your comment. Based on the previous question and explanation, it
appears that currently there are no clinically approved or failed drugs that target SREBPs and
their partners. This suggests that pursuing the development of new potential drugs and
conducting clinical studies in this area may hold promise.
5. Write down the best strategies to get protected from NAFLD progression in the
conclusion section.
Response: Thank you for bringing this issue to our attention. One of the key points of
significance in this review is to explore strategies for slowing or blocking the
progression of chronic liver diseases to severe stages. To enhance the conclusions
section, we have incorporated perceptive, criticism and potential strategies to address
this issue. Specifically, we suggest that a more precise and half-life SREBP inhibition
drug design could be a better strategy for preventing the progression of NAFLD. For
more information, please refer to lines 391-394 in the revised manuscript.

Round 2

Reviewer 3 Report

Comments and Suggestions for Authors

The Authors have addressed all my concerns and I have no further comments. 

As far as I am concerned, the manuscript is now acceptable to be published.